# ConPress: Learning Efficient Reasoning from Multi-Question Contextual Pressure

Jie Deng [1]   Shining Liang [2]   Jun Li [2]   Hongzhi Li [3]   Yutao Xie [2]

## Abstract

Large reasoning models (LRMs) typically solve reasoning-intensive tasks by generating long chain-of-thought (CoT) traces, leading to substantial inference overhead. We identify a reproducible inference-time phenomenon, termed **Self-Compression**: when multiple independent and answerable questions are presented within a single prompt, the model spontaneously produces shorter reasoning traces for each question. This phenomenon arises from *multi-question contextual pressure* during generation and consistently manifests across models and benchmarks. Building on this observation, we propose **ConPress** (Learning from **Con**textual **Press**ure), a lightweight self-supervised fine-tuning approach. ConPress constructs multi-question prompts to induce self-compression, samples the resulting model outputs, and parses and filters per-question traces to obtain concise yet correct reasoning trajectories. These trajectories are directly used for supervised fine-tuning, internalizing compressed reasoning behavior in single-question settings without external teachers, manual pruning, or reinforcement learning. With only 8k fine-tuning examples, ConPress reduces reasoning token usage by 59% on MATH500 and 33% on AIME25, while maintaining competitive accuracy.

## 1. Introduction

Large reasoning models (LRMs), such as OpenAI-O1 (Jaech et al., 2024), DeepSeek-R1 (Guo et al., 2025), and Qwen3 (Yang et al., 2025), have achieved strong performance on mathematics, coding, and other reasoning-intensive tasks by explicitly generating chain-of-thought (CoT) traces (Wei et al., 2022). While detailed reasoning can improve accuracy, it often contains redundant or unnecessary intermediate steps, a phenomenon commonly referred to as *overthinking* (Chen et al., 2024; Sui et al., 2025). Such verbosity substantially increases token usage, slows inference, and raises deployment costs, making reasoning efficiency a growing concern as LRMs are deployed at scale.

Prior work has explored reducing reasoning length primarily through supervised fine-tuning (SFT) and reinforcement learning (RL). SFT-based approaches typically rely on teacher models or curated pipelines to rewrite, prune, or distill long chain-of-thought traces into shorter ones, often requiring additional supervision or carefully designed heuristics (Jiang et al., 2025b; Yu et al., 2025; Arora & Zanette, 2025; Cheng et al., 2025b; Ma et al., 2025). These methods depend on external teachers or annotation processes, increasing training cost and complexity and limiting their scalability.

In parallel, reinforcement learning has emerged as a widely adopted strategy for controlling reasoning verbosity. Recent work incorporates token-level penalties, budget constraints, or reward shaping to encourage concise generations during RL training (Shao et al., 2024; Luo et al., 2025b; Team et al., 2025; Aggarwal & Welleck, 2025; Hou et al., 2025; Cheng et al., 2025a; Gao et al., 2025). Although powerful, RL-based approaches are often expensive, sensitive to reward design, and require substantial engineering effort to achieve stable training.

These limitations motivate a more fundamental question: *can large reasoning models naturally produce shorter reasoning traces, and under what conditions does this behavior emerge?*

To investigate this question empirically, we examine how LRMs behave when the structure of the input prompt varies. A consistent pattern emerges: when multiple independent and answerable questions are presented within a single prompt, the model produces shorter chain-of-thought traces for each question. This compression becomes evident as the prompt transitions from a single-question to a two-question setting. As the number of questions further increases, the extent of compression continues to grow while gradually

[1]University of Chinese Academy of Sciences, Beijing, China [2]Microsoft, Beijing, China [3]Tongji University, Shanghai, China. Correspondence to: Jie Deng <dengjie530@outlook.com>, Jun Li <junli1@microsoft.com>.

*Proceedings of the $43^{rd}$ International Conference on Machine Learning*, Seoul, South Korea. PMLR 306, 2026. Copyright 2026 by the author(s).

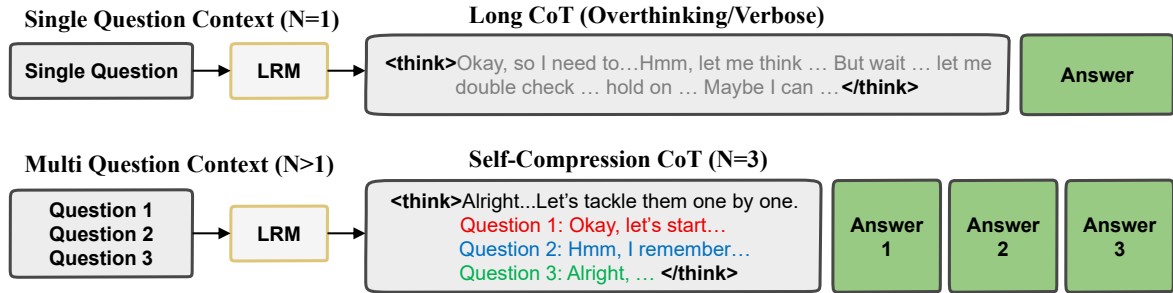

*Figure 1.* Illustration of single-question and multi-question decoding. By requiring multiple questions to be answered within a single generation, multi-question contexts introduce contextual pressure, which shortens the per-question reasoning traces.

stabilizing, forming a smooth and reproducible trend across models and reasoning tasks.

We interpret this behavior as a consequence of *multi-question contextual pressure*. In single-question prompts, the model operates in a context that implicitly favors extended elaboration on a single reasoning trajectory. When multiple independent questions are presented together, the prompt induces a different contextual state, in which several reasoning processes must be completed within a shared response. This contextual pressure alters the model's generation dynamics, biasing local continuations toward more concise reasoning paths that remove redundant intermediate steps while preserving the core inference structure. Importantly, this effect arises without explicit planning or resource allocation, reflecting a context-induced shift in continuation preferences during generation.

Building on the observation that multi-question contextual pressure induces systematic self-compression in reasoning, we introduce **ConPress**, a lightweight self-supervised fine-tuning framework that treats self-compressed reasoning traces as a reusable learning signal. ConPress leverages multi-question prompts as a mechanism for eliciting concise yet valid per-question reasoning trajectories, which reflect the model's compressed reasoning behavior under contextual pressure. By isolating these per-question traces and using them to supervise single-question fine-tuning, ConPress transfers this compressed reasoning behavior back to standard inference settings. Through this process, the model internalizes more token-efficient reasoning patterns without relying on external teachers, manual pruning heuristics, or reinforcement learning.

Across challenging benchmarks including MATH500, AIME, and AMC, ConPress achieves a **30–60%** reduction in reasoning token usage, accompanied by an accuracy–efficiency trade-off.

**Contributions.**

- We identify and systematically characterize a reproducible inference-time phenomenon, termed **self-**

**compression**, in which LRMs generate shorter per-question chain-of-thought traces when operating under multi-question contexts, without any explicit length constraints.

- We propose **ConPress**, a lightweight self-supervised fine-tuning framework that extracts and transfers self-compressed reasoning behavior from multi-question to single-question settings, without external teachers, manual pruning, or reinforcement learning.

- We empirically show that ConPress reduces reasoning token usage by **30–60%** on challenging benchmarks, exposing a clear accuracy–efficiency trade-off for token-efficient reasoning.

## 2. Self-Compression under Multi-Question Contextual Pressure

In this section, we study the decoding behavior of large reasoning models under multi-question contexts. Compared to single-question prompting, this setting introduces *contextual pressure* at inference time, as the model must complete multiple reasoning processes within a single generation. Under such pressure, we observe that the model systematically shortens its per-question reasoning traces. We refer to this phenomenon as *self-compression*. Figure 1 provides an intuitive illustration of this effect, which we analyze empirically in the remainder of this section across different prompt configurations, numbers of questions, and model families.

### 2.1. Problem Setting and Notation

**Single-question.** Given a single question $q$, a large reasoning model (LRM) produces an output consisting of a reasoning trace followed by a final response, which we denote as $\{\langle\texttt{think}\rangle r\langle/\texttt{think}\rangle o\}$ for models that explicitly mark reasoning spans. Here, $r$ denotes the reasoning trace and $o$ the corresponding model response. We define the reasoning length as $L = |r|$.

**Multi-question.** We consider a prompt containing $N$ in-

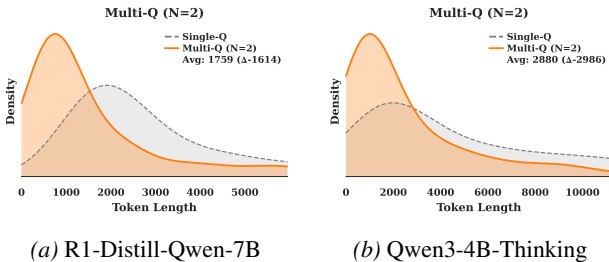

*Figure 2.* Distributions of per-question reasoning length under single-question ($N = 1$) and two-question ($N = 2$) prompting.

*Table 1.* Specificity analysis of self-compression under multi-question contexts. We compare the multi-question setting with control conditions that vary the difficulty of the additional question or the prompt structure without introducing an additional question.

| Condition | R1-Distill-Qwen-7B | | Qwen3-4B-Thinking | |
|---|---|---|---|---|
| | $\Delta L_1$ | $\rho_1$ (%) | $\Delta L_1$ | $\rho_1$ (%) |
| *Prompt modifications without an additional question* | | | | |
| Statement | −509 | 15.1 | −497 | 8.5 |
| Empty question | −582 | 17.3 | −857 | 14.6 |
| Concise instruction | −787 | 23.3 | −1442 | 24.6 |
| *Varying the difficulty of the second question* | | | | |
| Toy (1+1=?) | −1462 | 43.3 | −2895 | 49.3 |
| Easy | −1408 | 41.7 | −3006 | **51.2** |
| Medium | −1515 | 44.9 | −2771 | 47.2 |
| Hard | −1664 | **49.3** | −2695 | 45.9 |
| Random | −1606 | 47.6 | −2986 | 50.9 |

dependent questions, denoted by $Q = (q_1, \ldots, q_N)$, which must be answered within a single response. Given $Q$, the LRM produces an output consisting of reasoning traces $\{r_1, \ldots, r_N\}$ and corresponding responses $\{o_1, \ldots, o_N\}$, where $r_i$ and $o_i$ correspond to question $q_i$. The reasoning length for question $q_i$ under an $N$-question prompt is defined as $L_i^{(N)} = |r_i|$. We define the corresponding compression rate as $\rho_i^{(N)} = 1 - L_i^{(N)}/L_i^{(1)}$, where $L_i^{(1)}$ denotes the reasoning length for the same question under the single-question setting.

## 2.2. Self-Compression Phenomenon

We empirically examine the self-compression phenomenon under multi-question contexts, focusing on how per-question reasoning length varies with the structure of the input prompt. All experiments are conducted under fixed decoding conditions, with only the prompt composition varied. Unless otherwise specified, results are reported on two representative reasoning models, DeepSeek-R1-Distill-Qwen-7B (Guo et al., 2025) and Qwen3-4B-Thinking-2507 (Yang et al., 2025), using questions drawn from the MATH dataset (Hendrycks et al., 2021).

**Emergence at $N = 2$.** We compare single-question prompting ($N = 1$) with two-question prompting ($N = 2$), where two independent questions are answered within a single prompt. Figure 2 shows that even introducing a second question already results in a pronounced contraction of per-question reasoning length across both models, reflected by a systematic leftward shift of the reasoning-length distributions.

**Specificity to multi-question contexts.** Given the effectiveness of multi-question prompting, we investigate whether other prompt modifications can produce comparable effects on per-question reasoning length. To this end, we consider several alternative prompt designs that do not introduce an additional question, including adding a declarative statement, an empty question placeholder, or a concise instruction. As shown in Table 1, these prompt variants result in only limited reductions in reasoning length, and their effects are consistently much weaker than those observed under

multi-question contexts.

We then examine how the content of the additional question affects self-compression in multi-question settings. Surprisingly, we find that appending even a trivial arithmetic question (e.g., "1+1=?") after the target question already induces a substantial compression of the target reasoning trace, with nearly a 50% reduction in length. Moreover, varying the difficulty of the auxiliary question—from trivial to hard—leads to only modest differences in the resulting compression rate, which remains comparable across difficulty levels. This observation indicates that the pressure induced by multi-question contexts is largely structural in nature, and only weakly modulated by the difficulty of the auxiliary task.

**Scaling with the number of questions.** Figure 3 illustrates how self-compression evolves as the number of questions $N$ increases. We keep the target question fixed and vary $N$, measuring the reasoning length of the target question under different multi-question contexts. As shown in Figure 3a, both models exhibit a clear and monotonic decrease in reasoning length as $N$ increases. For R1-Distill-Qwen-7B, the compression ratio rises from 47.9% at $N = 2$ to 67.3% at $N = 8$, while for Qwen3-4B-Thinking it increases from 51.0% to 63.0 over the same range. This scaling trend indicates that self-compression intensifies as the strength of multi-question contextual constraints increases.

We further examine the effect of increasing $N$ on answer accuracy. Figure 3b reports relative accuracy under different multi-question settings. Although accuracy degrades gradually with larger $N$ for both models, Qwen3-4B-Thinking consistently exhibits greater robustness, whereas R1-Distill-Qwen-7B experiences a more pronounced decline.

Beyond average trends, Figures 3c and 3d provide distributional evidence. For both models, increasing $N$ leads to a systematic leftward shift in the reasoning-length distribu-

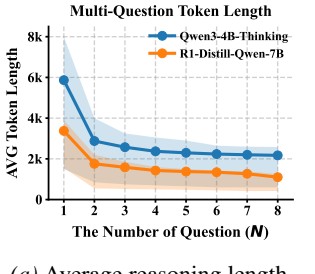
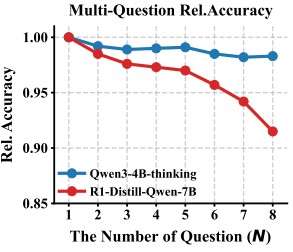

*(a)* Average reasoning length     *(b)* Relative accuracy

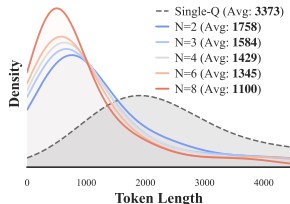
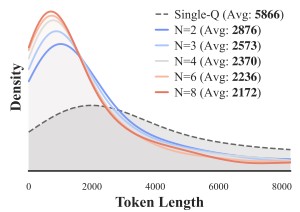

*(c)* Reasoning-length distributions for R1-Distill-Qwen-7B.     *(d)* Reasoning-length distributions for Qwen3-4B-Thinking.

*Figure 3.* Scaling of self-compression with the number of questions $N$. Top: average reasoning length and relative accuracy. Bottom: reasoning-length distributions across different $N$.

tions. This suggests that self-compression is a pervasive effect across instances, rather than being driven by a small number of outliers.

More details and additional experiments across datasets and model families are provided in Appendix A.

## 3. Method

Motivated by our empirical findings that multi-question contextual pressure induces systematically shorter reasoning traces (Section 2), we aim to transfer this concise reasoning behavior to standard single-question inference. To this end, we introduce **ConPress**, a lightweight self-supervised fine-tuning method that leverages the model's own self-compressed reasoning traces as supervision. ConPress follows a simple multi-to-single pipeline. We first elicit self-compressed reasoning traces by sampling the model under multi-question contexts. Since multi-question pressure may degrade answer accuracy, we apply rejection sampling to retain only correct reasoning trajectories. Finally, we distill this behavior into single-question inference via supervised fine-tuning.

### 3.1. Multi-Question Sampling

Let $\mathcal{Q}$ denote the set of single-question inputs. We sample $N$ independent questions $\{q_1, \ldots, q_N\} \sim \mathcal{Q}$ and pack them into a single prompt

$$P = \mathrm{Pack}(q_1, \ldots, q_N),$$

where $\mathrm{Pack}(\cdot)$ concatenates questions using a fixed neutral delimiter of the form `Question i:`. This prompt format matches the multi-question setting studied in Section 2, and introduces contextual pressure without imposing any explicit constraints on reasoning length or style.

Given the packed prompt $P$, the model generates per-question reasoning traces and predicted answers

$$\{(r_i^{(N)}, \hat{o}_i)\}_{i=1}^N \sim p_\theta(\cdot \mid P),$$

where $r_i^{(N)}$ and $\hat{o}_i$ denote the reasoning trace and predicted answer for question $q_i$ under an $N$-question context. These generations serve as the raw source of self-compressed reasoning trajectories.

### 3.2. Compressed Trace Extraction

From the sampled generations, we retain only those reasoning traces $r_i^{(N)}$ for which the predicted answer $\hat{o}_i$ matches the ground-truth answer $o_i$, and discard incorrect or malformed outputs. This rejection step is necessary to counteract the accuracy degradation observed under multi-question contexts, and ensures that only correct self-compressed reasoning trajectories are used as supervision.

Aggregating across prompts yields a dataset of concise and correct reasoning traces:

$$\mathcal{D}_{\mathrm{CP}} = \{(q_i, r_i^{(N)}, o_i)\}.$$

All supervision in $\mathcal{D}_{\mathrm{CP}}$ is produced entirely by the model itself, without external teacher models, manual rewriting, or heuristic compression rules.

### 3.3. Single-Question Transfer

We transfer the concise reasoning behavior captured in $\mathcal{D}_{\mathrm{CP}}$ to standard single-question inference via supervised fine-tuning. Given $\mathcal{D}_{\mathrm{CP}}$, the model is trained to reproduce the compressed reasoning trace for each question when conditioned on that question alone.

Formally, for each $(q_i, r_i^{(N)}) \in \mathcal{D}_{\mathrm{CP}}$, we minimize the token-level negative log-likelihood

$$\mathcal{L}_{\mathrm{SFT}}(\theta) = -\mathbb{E}_{(q_i, r_i^{(N)}) \sim \mathcal{D}_{\mathrm{CP}}} \sum_{t=1}^{|r_i^{(N)}|} \log p_\theta\left(r_{i,t}^{(N)} \mid q_i, r_{i,<t}^{(N)}\right).$$

Through this process, the model internalizes concise reasoning trajectories that were previously elicited only under multi-question contextual pressure, enabling more token-efficient reasoning during single-question inference.

*Table 2.* Accuracy and token usage across benchmarks. AVG only reports relative change $\Delta$. Original rows shown in gray.

| Model | MATH500 | | AIME25 | | GSM8K | | Olympiad | | AMC | | AVG | |
|---|---|---|---|---|---|---|---|---|---|---|---|---|
| | Acc | Tok. | Acc | Tok. | Acc | Tok. | Acc | Tok. | Acc | Tok. | $\Delta$Acc | $\Delta$Tok. |
| **Qwen3-4B-Thinking** | | | | | | | | | | | | |
| Original | 95.6 | 6634 | 72.5 | 21442 | 95.1 | 1509 | 73.3 | 14857 | 99.1 | 10772 | – | – |
| RFT$_{shortest}$ | **96.0** | 6062 | **72.9** | 21085 | **95.5** | 1367 | **72.6** | 14317 | **99.6** | 10137 | **+0.2** | -5.9% |
| DPO$_{shortest}$ | 95.2 | 5616 | 71.6 | 21372 | 94.8 | 1110 | 72.0 | 13434 | 97.5 | 9309 | -0.9 | -13.1% |
| **ConPress** | **96.0** | 2661 | 70.1 | 14258 | 95.0 | 729 | **72.6** | 8903 | 98.8 | 4482 | -0.6 | **-48.7%** |
| **R1-Distill-Qwen-7B** | | | | | | | | | | | | |
| Original | 91.6 | 3136 | 37.9 | 10643 | 90.2 | 993 | 58.7 | 6898 | 88.1 | 5346 | – | – |
| RFT$_{shortest}$ | **92.8** | 3148 | **41.7** | 10541 | **91.3** | 994 | 56.4 | 7025 | **87.5** | 5417 | **+0.6** | +0.5% |
| DPO$_{shortest}$ | 90.4 | 1918 | 32.1 | 8273 | 90.0 | 607 | 56.1 | 5385 | 86.5 | 3463 | -2.3 | -31.4% |
| LC-R1 | 87.6 | 1487 | 35.8 | 7339 | 86.8 | 433 | 58.1 | 4090 | 82.5 | 2862 | -3.1 | **-45.4%** |
| AdaptThink | 91.8 | 1550 | 34.2 | 9525 | 90.6 | 374 | 57.8 | 5312 | 85.9 | 3483 | -1.2 | -36.3% |
| **ConPress** | 92.4 | 1720 | 37.5 | 8698 | 91.2 | 547 | **58.2** | 5129 | 87.2 | 3459 | +0.0 | -33.9% |
| **R1-Distill-Qwen-1.5B** | | | | | | | | | | | | |
| Original | 81.2 | 4622 | 22.9 | 12176 | 82.6 | 2080 | 43.6 | 8866 | 67.8 | 7748 | – | – |
| RFT$_{shortest}$ | 82.4 | 4625 | 20.0 | 12362 | 83.4 | 2237 | 44.1 | 8625 | 67.8 | 7275 | -0.1 | +0.1% |
| DPO$_{shortest}$ | 82.0 | 3133 | 21.7 | 9926 | **85.4** | 1094 | 44.1 | 5732 | **68.7** | 5633 | **+0.8** | -32.2% |
| LC-R1 | 80.2 | 2313 | 21.6 | 7035 | 78.5 | 571 | 42.8 | 4545 | 66.2 | 3727 | -1.7 | -53.1% |
| ThinkPrune | **83.0** | 2587 | 20.4 | 7296 | 82.9 | 887 | **44.9** | 5094 | 65.0 | 4231 | -0.4 | -45.9% |
| AdaptThink | 80.4 | 1945 | 21.7 | 7534 | 83.2 | 490 | 42.8 | 4808 | 66.3 | 3027 | -0.7 | **-55.8%** |
| **ConPress** | 80.8 | 2255 | **22.5** | 8195 | 84.2 | 1095 | 43.0 | 5321 | 66.3 | 5442 | -0.2 | -40.2% |

## 4. Experiments

### 4.1. General Setup

We evaluate ConPress under a unified experimental setup covering data construction, training configuration, evaluation protocol, and baseline comparison. The same model is used both for sampling multi-question traces and for subsequent supervised fine-tuning.

**Data.** The training corpus is built from three sources: the MATH dataset, AIME problems prior to 2024, and the LIMO dataset (Ye et al., 2025), resulting in approximately 8k single-question items. To obtain concise chain-of-thought supervision, we apply multi-question sampling with $N = 3$. Sampling is performed with vLLM using a 32k context window, temperature 0.6, and top-$p$ 0.95. The 32k context length is chosen as a conservative upper bound to ensure that almost all multi-question rollouts are fully accommodated without truncation.

**Training.** We fine-tune three reasoning models: **Qwen3-4B-Thinking**, **R1-Distill-Qwen-7B**, and **R1-Distill-Qwen-1.5B**. Training uses standard negative log-likelihood optimization with a learning rate of $2 \times 10^{-5}$ and a batch size of 32. We enable sequence packing to maximize context utilization and train using the ms-swift framework (Zhao et al., 2024). Context parallelism and ZeRO-1 optimization

(Rasley et al., 2020) are applied to reduce memory overhead.

**Cost.** For the R1-Distill-Qwen-7B setting, ConPress generates about 24k raw multi-question rollouts (231M tokens), which are parsed into 51k single-question candidates with an average correctness rate of 84.9%. Selecting the shortest correct trace for each question yields about 8k final training samples (11M tokens), and the full sampling-plus-fine-tuning pipeline costs about 84 A100 GPU hours.

**Evaluation.** We evaluate on a diverse suite of math and reasoning benchmarks: MATH500 (Lightman et al., 2023), GSM8K (Cobbe et al., 2021), AIME25, OlympiadBench, and AMC. Qwen3-4B-Thinking is evaluated with a 32k generation limit, while R1-Distill-Qwen-7B and R1-Distill-Qwen-1.5B use a 16k limit. We report two metrics: (i) final-answer accuracy and (ii) the average number of generated tokens, which reflects reasoning efficiency. Following common practice, we report **Avg@8** for AIME25 and AMC due to their small size, and **Pass@1** for all other benchmarks.

**Baselines.** We compare ConPress with several approaches designed to reduce reasoning length:

- **RFT$_{shortest}$** (Rejection Fine-Tuning). For each question, multiple samples are generated and the shortest correct response is selected as the training target.

- **DPOshortest** (Direct Preference Optimization). For each question, both preference candidates are selected from correct responses: the shortest correct response is used as the chosen response, while the longest correct response is used as the rejected response. We follow a standard DPO setup with an additional 0.3-weighted NLL loss term for stability.

- **RL-based methods**. We include ThinkPrune (Hou et al., 2025), LC-R1 (Cheng et al., 2025b), and Adapt-Think (Zhang et al., 2025). For fairness, all baselines are evaluated by running publicly released checkpoints under our unified decoding configuration.

## 4.2. Main Results

**Substantial reduction in reasoning length.** As shown in Table 2, ConPress consistently achieves large reductions in chain-of-thought length across all models and benchmarks. On Qwen3-4B-Thinking, ConPress reduces average token usage by **48.7%**. For R1-Distill-Qwen-7B and R1-Distill-Qwen-1.5B, the average reductions are **33.9%** and **40.2%**, respectively. These gains are observed uniformly across arithmetic, competition math, and Olympiad-style benchmarks.

**Limited accuracy loss under aggressive compression.** Despite the substantial reduction in reasoning length, accuracy degradation remains small. On Qwen3-4B-Thinking, ConPress incurs an average accuracy drop of **0.6** points. On R1-Distill-Qwen-7B, average accuracy is preserved, while on R1-Distill-Qwen-1.5B the decrease is limited to **0.2** points. These results suggest that the compressed trajectories distilled from multi-question contexts retain the core reasoning steps required for correct problem solving.

**Comparison with existing compression methods.** Across all model sizes, ConPress provides a more favorable trade-off between accuracy and efficiency than prior approaches. RFTshortest yields only marginal reductions in token usage, while DPOshortest achieves moderate compression but often at the cost of noticeable accuracy degradation. RL-based methods such as LC-R1 and AdaptThink produce stronger compression, but with larger accuracy drops and substantially more complex training pipelines. In contrast, ConPress attains compression levels comparable to RL-based approaches while maintaining accuracy closer to the original models, using only standard supervised fine-tuning.

**Consistency across benchmarks and difficulty levels.** ConPress behaves robustly across benchmarks with varying difficulty. As illustrated in Figure 4, larger compression is observed on easier problems, while harder datasets such as AIME25 exhibit more moderate reductions accompanied by small accuracy drops. This pattern suggests that ConPress removes redundant reasoning more aggressively

*Table 3.* Comparison with direct test-time compression baselines on R1-Distill-Qwen-7B. MQ-N2/N3 directly pack multiple test questions into one prompt, while NoThinking suppresses explicit reasoning with an empty thinking prefix.

| Method | MATH500 | | GSM8K | | Avg. $\Delta$ | |
|---|---|---|---|---|---|---|
| | Acc. | Tok. | Acc. | Tok. | Acc. | Tok. |
| Original | 91.6 | 3136 | 90.2 | 993 | – | – |
| MQ-N2 | 66.2 | 2383 | 55.8 | 471 | -29.9 | -38.3% |
| MQ-N3 | 60.8 | 2271 | 58.4 | 388 | -31.3 | -44.3% |
| NoThinking | 79.4 | **779** | 87.0 | **299** | -7.7 | **-72.5%** |
| ConPress | **92.4** | 1720 | **91.2** | 547 | **+0.9** | -45.0% |

where possible, while preserving necessary computation on challenging problems.

**Why not use test-time compression directly?** A natural alternative is to apply compression-oriented prompting only at inference time. We evaluate multi-question prompting (MQ-N2/MQ-N3), which packs multiple test questions into one prompt, and NoThinking, which suppresses explicit reasoning with an empty thinking prefix. As shown in Table 3, these test-time shortcuts reduce token usage but substantially compromise accuracy. MQ-N3 reduces the average relative token usage by 44.3% over MATH500 and GSM8K, but drops accuracy by 31.3 points on average. NoThinking achieves the strongest compression (72.5% token reduction), yet still suffers a 7.7-point average accuracy drop. In contrast, ConPress preserves accuracy, improving average accuracy by 0.9 points while reducing token usage by 45.0%. Thus, compression is more reliable when distilled from multi-question contexts into the standard single-question setting, rather than imposed directly at inference time.

## 4.3. Ablation Study

**The Effect of $N$.** In Section 2, we observed that increasing the number of questions in the prompt induces stronger self-compression at inference time, producing shorter per-question reasoning traces. Here we explore whether the same scaling tendency carries over to *training* in ConPress: i.e., whether sampling with larger $N$ yields systematically more compressed supervision and leads to more token-efficient models after fine-tuning.

Table 4 shows a consistent trend in reasoning cost: larger $N$ produces shorter post-training traces on both MATH500 and GSM8K. This establishes $N$ as a practical control knob for the compression strength of ConPress, allowing us to trade supervision conciseness for performance stability. At the same time, the marginal token savings diminish as $N$ increases: the gain from $N = 2$ to $N = 3$ is substantial ($-35.7\% \rightarrow -45.0\%$), whereas further increasing $N$ yields relatively modest improvements (e.g., $-49.0\%$ at $N = 4$ and $-49.4\%$ at $N = 6$).

*Table 4.* Effect of $N$ in ConPress. We report accuracy (**Acc.**) and average token usage (**Tok.**) on MATH500 and GSM8K.

| | MATH500 | | GSM8K | | $\Delta$Avg | |
|---|---|---|---|---|---|---|
| **Model Setting** | Acc. | Tok. | Acc. | Tok. | Acc. | Tok. |
| Original (Baseline) | 91.6 | 3136 | 90.2 | 993 | – | – |
| ConPress ($N=2$) | 90.8 | 2003 | 91.6 | 642 | +0.3 | -35.7% |
| ConPress ($N=3$) | **92.4** | 1720 | 91.2 | 547 | **+0.9** | -45.0% |
| ConPress ($N=4$) | 91.4 | 1655 | **91.8** | 488 | +0.7 | -49.0% |
| ConPress ($N=6$) | 89.8 | 1702 | 91.2 | 466 | -0.4 | -49.4% |
| ConPress ($N=8$) | 91.0 | **1489** | 91.2 | **450** | +0.2 | **-53.6%** |

*Table 5.* Effect of sampling position $k$ in ConPress. **Position** indicates training the model using the $k$-th question.

| | MATH500 | | GSM8K | | $\Delta$Avg | |
|---|---|---|---|---|---|---|
| **Position** | Acc. | Tok. | Acc. | Tok. | Acc. | Tok. |
| Original (Baseline) | 91.6 | 3136 | 90.2 | 993 | – | – |
| Position 1 | 90.4 | **1757** | **92.8** | 562 | +0.7 | -49.3% |
| Position 2 | 90.8 | 1902 | 92.0 | 634 | +0.5 | -40.8% |
| Position 3 | **91.0** | 1909 | 91.8 | 708 | +0.5 | -37.8% |

Accuracy, however, does not follow a simple monotonic pattern across $N$. To preserve accuracy while still obtaining meaningful compression, moderate values such as $N=3$ or $N=4$ provide a favorable trade-off in our setting. We adopt $N=3$ as the default: it achieves strong token reduction ($-45.0\%$) while yielding the largest average accuracy improvement across the two benchmarks ($+0.9$).

**The Effect of Sampling Position.** Beyond the number of questions $N$, an additional design consideration in ConPress is whether compressed supervision must be sampled from a specific position within a multi-question prompt. If effective compression were strongly tied to a particular position, ConPress would require rigid prompt layouts and limit flexibility during data collection.

To examine this effect, we train models using trajectories sampled exclusively from a single position and evaluate them on standard single-question benchmarks. As shown in Table 5, models trained from different positions achieve comparable accuracy after fine-tuning. This suggests that, within our experimental setting, ConPress does not critically depend on a fixed sampling position to obtain effective compressed supervision. At the same time, trajectories sampled from earlier positions tend to exhibit stronger compression signals, leading to shorter reasoning traces after training (e.g., $-49.3\%$ vs. $-37.8\%$ in $\Delta$Tok.).

Based on these observations, we do not restrict ConPress training to a specific position. Instead, we allow questions to appear at arbitrary positions within multi-question prompts and retain all resulting single-question trajectories after filtering. Under this strategy, a single multi-question genera-

*Table 6.* Stage-wise reasoning efficiency for R1-Distill-Qwen-7B before and after ConPress fine-tuning. We report pre-solution thinking tokens (**Pre**), total thinking tokens (**Tok**), and reasoning efficiency ratio (**Ratio**).

| | Reasoning Statistics | | | | | |
|---|---|---|---|---|---|---|
| | Original | | | ConPress | | |
| **Dataset** | Pre $\downarrow$ | Tok $\downarrow$ | Ratio $\uparrow$ | Pre $\downarrow$ | Tok $\downarrow$ | Ratio $\uparrow$ |
| AIME25 | 7791 | 8562 | **0.91** | 5480 | 6089 | 0.90 |
| AMC | 2812 | 4135 | 0.68 | 2074 | 2529 | **0.82** |
| GSM8K | 496 | 870 | 0.57 | 266 | 444 | **0.60** |
| MATH500 | 1681 | 3001 | 0.56 | 1038 | 1403 | **0.74** |

tion can yield multiple usable training examples, substantially improving sampling efficiency in our data collection pipeline.

## 5. Analysis

### 5.1. Compression Across Difficulty Levels

We examine the behavior of ConPress across problem difficulty using the five-level partition of the MATH500 benchmark. Figure 4 reports the average reasoning length and accuracy before and after ConPress fine-tuning for Qwen3-4B-Thinking and R1-Distill-Qwen-7B. Across both models, ConPress consistently reduces chain-of-thought length at all difficulty levels. The extent of compression, however, varies with difficulty: larger reductions are observed at lower levels, while higher levels exhibit more moderate decreases. This pattern is consistent across the two models and indicates that compression strength is not uniform across difficulty.

Accuracy remains largely stable across difficulty levels. In particular, harder problems exhibit more moderate compression while accuracy remains largely stable. A similar tendency is also observed across benchmarks: as shown in Table 2, compression on the more challenging AIME25 benchmark is substantially smaller than on MATH500. Taken together, these results show that ConPress tends to achieve larger token reductions where strong compression is attainable, while exhibiting more conservative behavior on harder problems, thereby preserving accuracy.

### 5.2. Reasoning Behavior Analysis

**Reasoning Efficiency.** We analyze how computation is distributed along the reasoning trace. For each question, let $C^{\mathrm{pre}}$ denote the number of thinking tokens generated before the first correct answer, and let $C^{\mathrm{tot}}$ denote the total number of thinking tokens. We define the reasoning efficiency ratio as $\eta = C^{\mathrm{pre}}/C^{\mathrm{tot}}$, and report its dataset-level average.

Table 6 analyzes how compression under ConPress is dis-

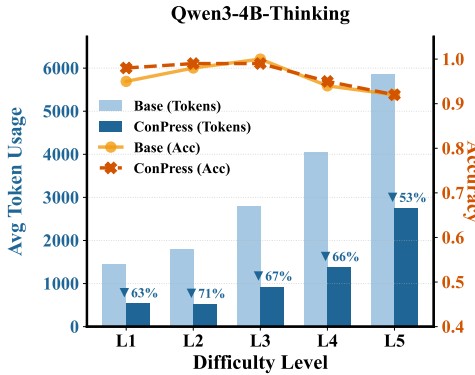 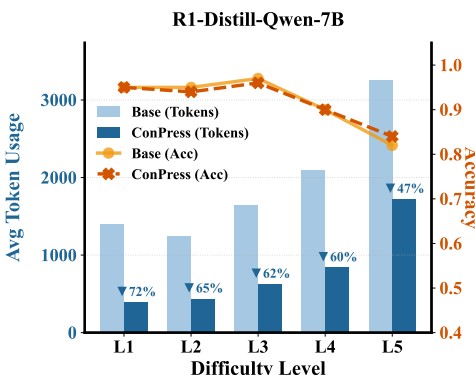

*Figure 4.* Effects of ConPress across difficulty levels on MATH500. ConPress consistently reduces reasoning length across all levels while largely preserving accuracy.

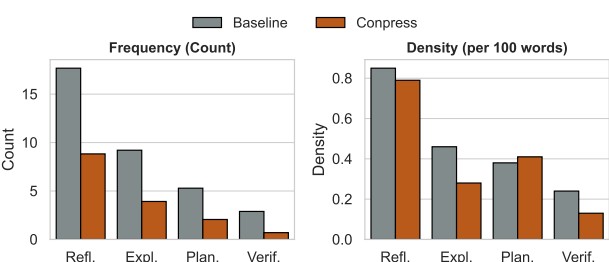

*Figure 5.* Distribution of reasoning behaviors before and after Con-Press training, shown in terms of frequency (left) and normalized density per 100 words (right).

*Table 7.* Generalization performance on the out-of-distribution MMLU-STEM benchmark. ConPress achieves substantial token reduction with only minor accuracy degradation.

| Model | Acc | Tokens | $\Delta$Acc | $\Delta$Tok |
|---|---|---|---|---|
| Qwen3-4B-Original | 93.3 | 2862 | – | – |
| Qwen3-4B-ConPress | 92.9 | 1816 | $-0.4$ | $-36.6\%$ |
| R1-7B-Original | 83.5 | 1548 | – | – |
| R1-7B-ConPress | 82.4 | 1035 | $-1.1$ | $-33.1\%$ |

### 5.3. Out-of-Distribution Evaluation

We evaluate ConPress on the MMLU-STEM subset to assess its generalization beyond mathematical tasks. MMLU-STEM spans nineteen science and engineering subjects and evaluates broad factual and reasoning ability. As shown in Table 7, ConPress substantially reduces inference token usage on this out-of-distribution benchmark while incurring only minor accuracy degradation. Specifically, Qwen3-4B-Thinking achieves a 36.6% reduction in tokens with a 0.4-point decrease in accuracy, and R1-Distill-Qwen-7B achieves a 33.1% reduction with a 1.1-point decrease. These results indicate that the compression effect of ConPress extends beyond mathematics to broader reasoning domains.

## 6. Related Work

### 6.1. Multi-Question Prompting.

Multi-question prompting has been explored mainly as an input construction or evaluation setting for large language models. Batch prompting groups multiple independent data samples into a single prompt for joint processing (Cheng et al., 2023; Lin et al., 2023), while several benchmarks and empirical studies investigate model robustness and consistency under multi-question or long-context inputs (Liu et al., 2024; Laskar et al., 2023; Son et al., 2024; Wang et al., 2025). REST (Pan et al., 2025) packs multiple reasoning-intensive problems into one prompt for evaluation, and

tributed along the reasoning trace. By explicitly reporting the estimated number of pre-solution thinking tokens, we observe that ConPress substantially compresses the solution-search stage itself across all benchmarks. At the same time, the consistent increase in the efficiency ratio $\eta$ indicates a non-uniform compression pattern, in which post-solution reasoning is reduced more aggressively than pre-solution reasoning. These results show that ConPress compresses both the solving and verification stages of reasoning, while preferentially suppressing extended post-solution verification and redundant continuation, which constitute a major source of overthinking in large reasoning models.

**Reasoning Behavior.** Thinking tokens are grouped by indicative lexical patterns into *planning*, *exploration*, *verification*, and *reflection* (e.g., "first", "what if", "check", "wait"). While the frequency of all behaviors decreases under Con-Press due to overall compression, the normalized density reveals a selective effect: exploration and verification are substantially reduced, whereas planning remains largely unchanged. This suggests that ConPress places a stronger emphasis on reducing overthinking-related behaviors, without eliminating solution-critical reasoning components.

MathFusion (Pei et al., 2025) creates new problems by fusing related math questions.

### 6.2. Efficient Reasoning in LRMs

A growing body of work aims to improve the efficiency of large reasoning models by mitigating unnecessarily long chain-of-thought traces. Most existing approaches rely on explicit training-time regulation of reasoning length. In supervised settings, models are trained to produce shorter or more concise rationales via pruning, rewriting, or reasoning-style control, often with external supervision or auxiliary models (Jiang et al., 2025b; Yu et al., 2025; Qiao et al., 2025). Another line of work employs online reinforcement learning, where generation length is optimized through reward shaping mechanisms such as token-budget constraints or length-aware penalties (Team et al., 2025; Arora & Zanette, 2025; Luo et al., 2025b; Cheng et al., 2025b; Hou et al., 2025; Yi et al., 2025). Adaptive or hybrid reasoning approaches train models to dynamically switch between longer and shorter reasoning depending on the input, balancing accuracy and efficiency (Zhang et al., 2025; Jiang et al., 2025a; Luo et al., 2025a; Fang et al., 2025).

### 7. Conclusion

We identify a reproducible self-compression phenomenon in LRMs, whereby multi-question prompts introduce contextual pressure that naturally shortens per-question reasoning traces during inference. Building on this observation, we propose ConPress, a lightweight training framework that leverages multi-question sampling and correctness-based filtering to extract concise yet valid reasoning trajectories and distill this behavior through standard supervised fine-tuning. Across multiple models and benchmarks, ConPress consistently reduces chain-of-thought length while maintaining competitive accuracy. These results show that self-compressed reasoning behaviors induced at inference time can be systematically internalized by the model itself, enabling more token-efficient reasoning without explicit length constraints, external teachers, or reinforcement learning.

### Impact Statement

This paper presents a simple and efficient training method for improving the efficiency of reasoning in large reasoning models, contributing to research on efficient reasoning. By reducing redundant reasoning tokens, the proposed approach lowers inference-time computational cost. We do not identify specific societal impacts that require separate discussion beyond those generally associated with large language models.

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

# A. More Analysis of Self-Compression

## A.1. Universality of Self-Compression Across Data Distributions

To verify whether the self-compression phenomenon is universally present across different data distributions beyond the primary evaluation set, we conducted extensive experiments using the **R1-Distill-Qwen-7B** model. We selected three distinct benchmarks representing diverse reasoning tasks, along with a mixture setting:

- **GPQA** (Google-Proof Q&A) (Rein et al., 2023): A challenging dataset consisting of graduate-level questions in biology, physics, and chemistry, serving as a proxy for high-level scientific reasoning.

- **MBPP** (Mostly Basic Python Problems) (Austin et al., 2021): A benchmark focusing on code generation capabilities, requiring logical synthesis and syntax correctness.

- **MATH500** (Hendrycks et al., 2021): A subset of the MATH benchmark, curated to evaluate multi-step mathematical reasoning.

- **Mixture**: A heterogeneous dataset constructed by interleaving samples from the three benchmarks above within the same context window.

As visualized in Figure 6, the self-compression phenomenon is consistently reproducible across all tested distributions. We observe two distinct patterns as the number of sequential problems ($N$) increases:

**Consistent Leftward Shift and Token Reduction.** The distribution of thinking tokens exhibits a significant leftward shift across all domains, indicating a universal reduction in computation cost. For scientific reasoning (GPQA), the average thinking tokens ($\mu$) decrease steadily from 4852 at $N = 1$ to 2294 at $N = 4$. This compression is even more pronounced in code generation (MBPP), where the mean drops drastically from 2353 to 697, retaining only about 30% of the original length. Mathematical reasoning (MATH500) follows a similar trajectory, reducing from 2286 to 1124.

**Robustness in Mixed Contexts.** Crucially, this mechanism holds even in the *Mixture* setting. Despite the complexity of switching between diverse reasoning types within a single window, the model compresses the reasoning process effectively, with the mean token count dropping from 2863 to 1337. This suggests that self-compression is an intrinsic capability of the model, independent of domain homogeneity.

**Distributional Sharpening.** Beyond the reduction in length, the shape of the distributions evolves from a broad dispersion at $N = 1$ to a sharper, more concentrated peak at

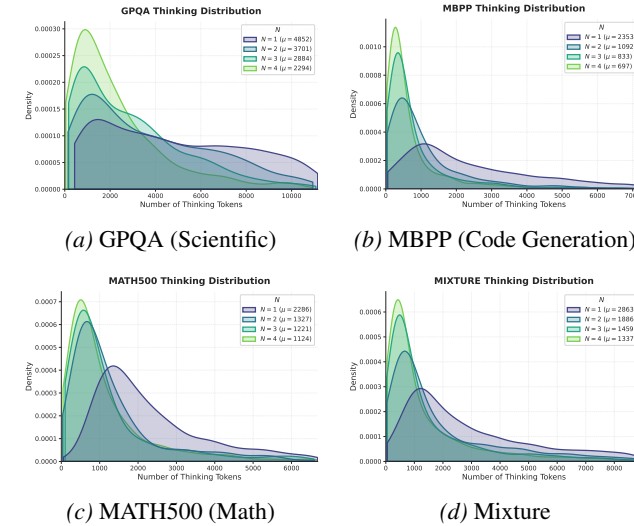

*(a)* GPQA (Scientific)  *(b)* MBPP (Code Generation)

*(c)* MATH500 (Math)  *(d)* Mixture

*Figure 6.* **Evolution of thinking token distributions across diverse benchmarks.** As the number of questions $N$ increases from 1 to 4, all benchmarks exhibit a consistent leftward shift in thinking-token distributions and a sharpening of density peaks, indicating systematic self-compression across scientific reasoning (GPQA), code generation (MBPP), mathematical reasoning (MATH500), and heterogeneous mixed contexts.

$N = 4$. This "peaking" effect implies that as the context pressure increases, the model's reasoning path becomes not only shorter but also more stable and deterministic.

## A.2. More Models

To further verify that the self-compression phenomenon induced by multi-question contextual pressure is not specific to a single model family, we evaluate several additional open-source large reasoning models (LRMs) using the same math-based multi-question prompt construction. These models differ in backbone architectures, parameter scales, and reasoning-oriented training designs, enabling a broader examination of the phenomenon.

We consider the following models and their corresponding reasoning annotation formats:

- **DeepSeek-R1-Distill-Llama-8B** (Guo et al., 2025) and **Qwen3-30B-A3B** (Yang et al., 2025): reasoning traces are explicitly enclosed within `<think>` ··· `</think>`.

- **Seed-OSS-36B** (Team, 2025): reasoning traces are marked using `<seed:think>` ··· `</seed:think>`.

- **ERNIE-4.5-21B-A3B-Thinking** (Baidu-ERNIE-Team, 2025): reasoning traces are enclosed by `<think>` ··· `</think>`, followed by a separate `<response>` ··· `</response>` block.

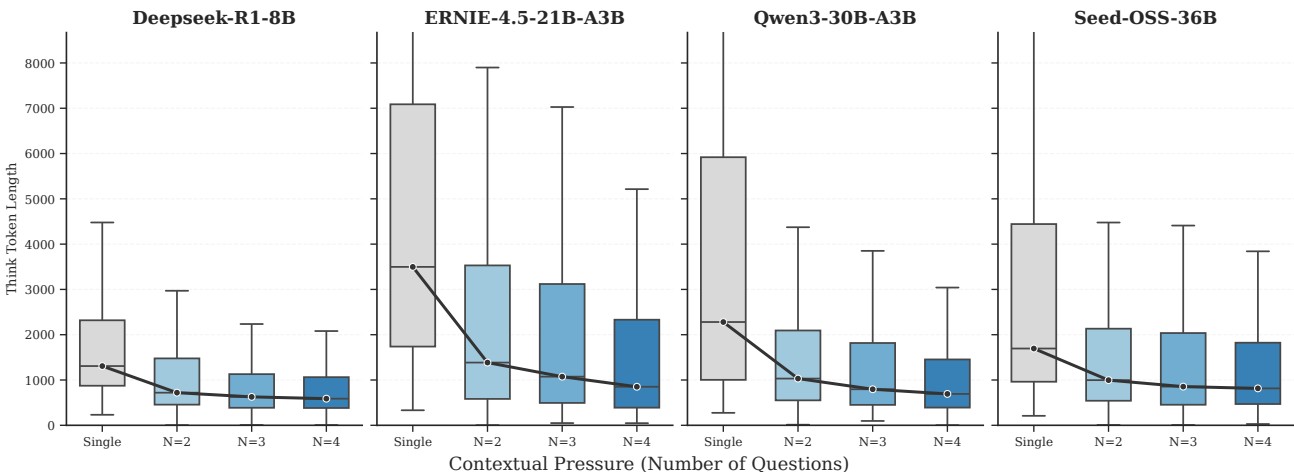

*Figure 7.* Distribution of reasoning token lengths across different models under single-question and multi-question settings. Each subfigure reports the token length of extracted reasoning spans as the number of questions increases from Single to $N=4$. All models exhibit a substantial reduction in reasoning length under multi-question contextual pressure.

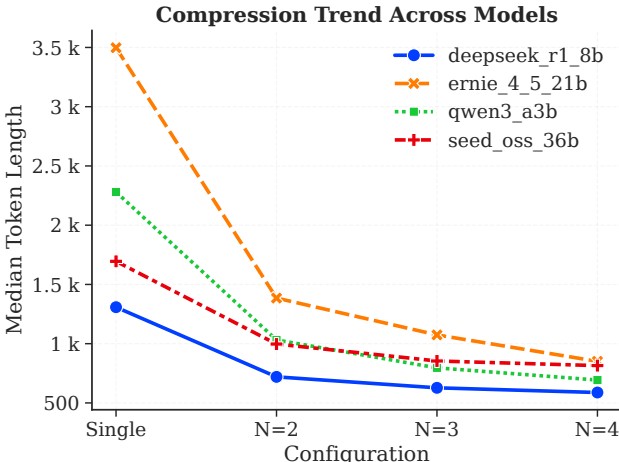

*Figure 8.* Median reasoning token length as a function of the number of questions. Compression trends are consistent across models, with the largest reduction typically occurring between the single-question and $N=2$ settings.

Model-specific regular expressions are applied accordingly to extract per-question reasoning segments, following the same parsing and filtering rules used throughout the paper.

Figure 7 presents the full distribution of reasoning token lengths for each model. Despite large differences in absolute reasoning length under single-question prompts, all models demonstrate a consistent downward shift in token usage as the number of questions increases, indicating that self-compression emerges robustly across architectures.

Figure 8 summarizes this effect by plotting the median reasoning length for each model. While the magnitude of compression varies, the monotonic decreasing trend with respect to the number of questions is consistent, further supporting

the generality of multi-question–induced self-compression across diverse LRMs.

### A.3. Case Study for Self-Compression

To provide an intuitive illustration of contextual-pressure-induced self-compression, we present an extreme toy example using **R1-Distill-7B**. We compare the model's reasoning behavior under a single-question prompt and a multi-question prompt containing two trivial arithmetic queries. As shown in Box A.1, when only a single question is given, the model exhibits clear overthinking behavior: the response contains extensive narration, repeated verification, and multiple forms of low-level reflection, despite the simplicity of the task. Even for a question as elementary as *"1+1=?"*, the model allocates a large number of tokens before committing to an answer.

In contrast, under the multi-question setting shown in Box A.2, the model's reasoning becomes markedly compressed. The intermediate thinking is reduced to a minimal form, and redundant reflection and double-checking processes are spontaneously removed. Importantly, this compression is not enforced by explicit length constraints or instructions. Instead, the presence of multiple questions in the same context appears to reshape the model's internal reasoning strategy, leading it to streamline the path to each answer and avoid allocating tokens to low-value deliberation.

---

**Single-question setting**

**Input:**
Question: 1+1=?

**Model output:** `<think>`

---

Okay, so I need to figure out what 1 plus 1 equals. Hmm, let me think. I remember from when I was a kid that addition is like putting things together. So if I have one apple and someone gives me another apple, how many apples do I have? That should be two apples, right? So 1 plus 1 equals 2.

But wait, maybe I should double-check that. Sometimes when I'm learning something new, I second-guess myself. Let me visualize it. Imagine my hand, that's one finger up, and if I raise another finger, how many do I have? Two fingers up. Yep, that's the same as 1 plus 1.

Is there another way to think about it? Maybe using objects. If I have one toy car and I get another toy car, how many toy cars do I have? That's two toy cars. So, again, 1 plus 1 equals 2.

What about using my fingers? If I put up one finger on my left hand and one finger on my right hand, how many fingers am I showing? That's two fingers. So that's another way to see that 1 plus 1 is 2.

Maybe I can think about it on a number line. Starting at 1 and then moving one step forward, where do I land? I land on 2. So that also shows that 1 plus 1 is 2.

Wait, is there a mathematical property that defines this? I think addition is commutative, which means the order doesn't matter. So 1 plus 1 is the same as 1 plus 1, which is still 2.

Is there a more formal way to represent this? In math, we use the plus sign '+' to denote addition. So, 1 + 1. When you add these two numbers units together, you get 2. So, 1 + 1 = 2

I think I've confirmed it through different methods: using fingers, objects, number lines, and the properties of addition. All of them lead me to the conclusion that 1 plus 1 equals 2. So, I'm pretty confident that the answer is correct. `</think>`

---

**Multi-question setting**

**Input:**
Question 1: 1+1=?
Question 2: 1*1=?

**Model output:** `<think>`
First, I'll tackle the first question, which is 1 plus 1. ***When I add 1 and 1 together, the result is 2.***
Next, for the second question, I'll solve 1 multiplied by 1. Multiplying 1 by itself also gives me 1. `</think>`

# B. Experiment Details

### B.1. Sampling and Parsing

We construct multi-question prompts by randomly grouping questions from the same dataset, ensuring that no prompt contains duplicated questions. All prompts are organized using neutral separators and explicit identifiers in the form "`Question 1: ...\n\n Question 2: ...\n\n Question 3: ...`", which facilitates both generation and downstream parsing. For each multi-question prompt, we perform multiple independent samplings to obtain more stable reasoning trajectories. Each prompt is sampled eight times, and all generated outputs are retained for subsequent processing.

During parsing, we extract the reasoning traces enclosed within the `<think>` block together with the corresponding final answers. The explicit question identifiers (e.g., `Question` $i$) serve as the primary anchors to segment the output into per-question reasoning blocks. Due to differences in response structure across models, minor model-specific parsing heuristics are applied when necessary. For example, Qwen3-series models often use explicit separators (e.g., "—") between questions, while DeepSeek-R1-style models frequently introduce characteristic discourse tokens (e.g., *"Okay"*, *"Hmm"*, *"Alright"*) at the beginning of each reasoning segment, which are used as auxiliary cues.

To ensure the correctness of compressed reasoning trajectories, we apply an automatic filtering step after parsing. For each question, we extract the boxed final answer from the model output and compare it against the ground-truth answer. We employ task-specific automatic verifiers, including numerical equivalence checks and symbolic formula matching, to determine correctness. Only samples that produce correct answers are retained, while incorrect responses are discarded. This filtering step ensures that the compressed trajectories used for training preserve solution correctness.

### B.2. Training Details

All models are fine-tuned using standard next-token prediction with negative log-likelihood loss. No auxiliary objectives, explicit length regularization, reinforcement learning, or preference optimization techniques are introduced during training. The training data consists of compressed reasoning trajectories that are filtered for correctness. Our experiments are conducted on a cluster of $8\times$ NVIDIA A100 GPUs with 80GB memory per device. We employ sequence parallelism with a parallel size of 2 to support long-context training efficiently. For DeepSeek-R1-Qwen, the maximum sequence length is set to 16,384 tokens, while for Qwen3, the maximum sequence length is increased to 32,768 tokens to accommodate its longer reasoning traces. Training is performed for 3 epochs with a learning rate of $2 \times 10^{-5}$ and a linear warmup ratio of 0.05. Samples originating from different question positions within multi-question prompts are mixed uniformly, and no position-specific weighting or curriculum strategy is applied.

From a training perspective, ConPress is lightweight and self-contained. It does not rely on external teacher models, handcrafted pruning rules, or reinforcement learning pipelines. Since the supervision is generated by the model itself, fine-tuning can be performed with a relatively small number of training examples and minimal additional engineering. In practice, we observe stable optimization behavior across all settings, and all models converge smoothly under standard supervised fine-tuning configurations.

## B.3. Evaluation Details

**Evaluation Setup.** All evaluations are conducted using `vLLM` version 0.7.3. Our evaluation pipeline is adapted from the official Qwen2.5-Math repository, with additional modifications to support multi-model inference, prompt customization, and large-scale batch evaluation. We perform evaluation using sampling with a temperature of 0.6 and top-$p$ of 0.95, following the official recommendations for DeepSeek-R1-style reasoning models.

**Prompt Templates.** Different models require different prompt formats to elicit chain-of-thought reasoning and produce properly formatted final answers. We adopt model-specific prompt templates, all of which instruct the model to reason step by step and place the final answer within `\boxed{}`. Representative templates are shown below.

---
**Qwen3 Prompt Template**

```
<|im_start|>system
Please reason step by step, and put your
final answer within \boxed{}.  <|im_end|>
<|im_start|>user
{input}<|im_end|>
<|im_start|>assistant
```
---
**R1-Distill Prompt Template**

```
<|begin_of_sentence|>
Please reason step by step, and put your
final answer within \boxed{}.
<|User|>{input}
<|Assistant|>
```
---

During evaluation, model outputs are parsed to extract the final boxed answer, which is then compared against the ground-truth solution using task-specific verifiers. The same evaluation protocol is applied consistently across all models and benchmarks.

# C. Case Study for ConPress

Boxes C.1 and C.2 show a concrete case study illustrating how ConPress changes the reasoning behavior of the model. In the original (Box C.1), the model reaches the correct answer but produces a lengthy reasoning trace, with repeated explanations, exhaustive enumeration, and multiple rounds of verification. After fine-tuning with ConPress (Box C.2), the model follows the same core solution strategy based on prime factorization, but expresses it in a much more compact form.

The ConPress-trained model directly focuses on the essential reasoning steps required to solve the problem, avoiding unnecessary elaboration. At the same time, it still includes brief consistency checks, such as validating the divisor count, in a concise and accurate manner. Overall, this case illustrates how ConPress yields substantially shorter

reasoning traces while preserving correctness and logical completeness.

---
**Original R1-7B**

**INPUT:** How many positive whole-number divisors does 196 have?
**OUTPUT:** `<think>`
Okay, so I have to figure out how many positive whole-number divisors 196 has. Hmm, divisors... that means numbers that can divide 196 without leaving a remainder. Let me think about how to approach this.

First, maybe I should start by factoring 196 into its prime factors. I remember that prime factorization is a good way to find the number of divisors. So, let me try that.

Alright, 196. I know that 196 is an even number because it ends with a 6, so it's divisible by 2. Let me divide it by 2. $196 \div 2$ is 98. Okay, so 2 is a prime factor, and now I have 98 left.

98 is also even, so I can divide by 2 again. $98 \div 2$ is 49. So far, my prime factors are $2 \times 2 \times$ something. Now, 49 is left. Hmm, 49 is a square number, right? It's 7 squared. So, $7 \times 7$ is 49. Therefore, the prime factors are 2, 2, 7, and 7.

Let me write that down: $196 = 2^2 \times 7^2$. So, that's the prime factorization.

Now, I remember there's a formula to find the number of divisors once you have the prime factors. If a number N can be expressed as $N = p_1^a \times p_2^b \times ... \times p_n^k$, where $p_1$, $p_2$, ..., $p_n$ are prime numbers, then the number of positive divisors is $(a+1)(b+1)...(k+1)$.

Let me make sure I remember that correctly. So, for each exponent in the prime factorization, you add 1 and then multiply them all together. That gives the total number of divisors.

So, applying that to 196. The exponents are 2 and 2 for the primes 2 and 7, respectively. So, adding 1 to each exponent gives me $(2+1)$ and $(2+1)$. Multiplying those together: $3 \times 3 = 9$.

Wait, so does that mean 196 has 9 positive divisors? Let me verify that by listing them out to be sure I didn't make a mistake.

Starting with 1, since 1 is a divisor of every number. Then 2, because it's even. 4, since $2 \times 2$ is 4 and 4 divides 196 (because $196 \div 4$ is 49). Next, 7, because $7 \times 7$ is 49, and 49 divides 196. Then, 14, because $2 \times 7$ is 14, and $14 \times 14$ is 196, so 14 is also a divisor.

Wait, hold on. Let me list them properly:
1 (since $1 \times 196 = 196$) 2 (since $2 \times 98 = 196$) 4 (since $4 \times 49 = 196$) 7 (since $7 \times 28 = 196$) 14 (since $14 \times 14 = 196$) 28 (since $14 \times 14$, but wait, I already had 14) Wait, maybe I should list them in pairs to avoid missing any.

Let me pair the divisors:
1 and 196 (because $1 \times 196 = 196$) 2 and 98 (because $2 \times 98 = 196$) 4 and 49 (because $4 \times 49 = 196$) 7 and 28 (because $7 \times 28 = 196$) 14 and 14 (because $14 \times 14 = 196$)

So, each pair multiplies to 196, so that gives me 1, 2, 4, 7, 14, 28, 98, and 196. Let me count them: 1, 2, 3, 4, 5, 6, 7, 8, 9. Yep, that's nine divisors. So, that matches the formula I used earlier.

So, using the formula, it's $(2+1)(2+1) = 9$, which is the same as listing them out. So, that seems correct.

Just to make sure, let me think if there are any other divisors I might have missed. Let me see:
- 3: $196 \div 3$ is about 65.333... which isn't a whole number, so

---

3 isn't a divisor. - 5: $196 \div 5$ is 39.2, not a whole number. - 6: $196 \div 6$ is about 32.666..., not a whole number. - 8: $196 \div 8$ is 24.5, not a whole number. - 9: $196 \div 9$ is approximately 21.777..., not a whole number. - 10: $196 \div 10$ is 19.6, not a whole number. - 11: $196 \div 11$ is approximately 17.818..., not a whole number. - 12: $196 \div 12$ is about 16.333..., not a whole number. - 13: $196 \div 13$ is approximately 15.076..., not a whole number.

So, none of those are whole numbers, which means 196 doesn't have any other divisors besides the ones I listed earlier. Therefore, 9 divisors is correct.

Wait, just to be thorough, I can list all the possible combinations of the prime factors to see if I get the same list.

The prime factors are 2² × 7². So, the exponents for 2 can be 0, 1, or 2, and for 7 can be 0, 1, or 2.

So, let's list all combinations:

- $2^0 \times 7^0 = 1$ - $2^0 \times 7^1 = 7$ - $2^0 \times 7^2 = 49$ - $2^1 \times 7^0 = 2$ - $2^1 \times 7^1 = 14$ - $2^1 \times 7^2 = 98$ - $2^2 \times 7^0 = 4$ - $2^2 \times 7^1 = 28$ - $2^2 \times 7^2 = 196$

So, that gives the same list as before: 1, 2, 4, 7, 14, 28, 49, 98, 196. So, that's nine divisors.

Therefore, both methods confirm that 196 has 9 positive whole-number divisors.

I think that's solid. I don't see any mistakes in my reasoning. I started by prime factorizing, applied the formula, and then double-checked by listing all possible combinations of the prime factors. Both methods led me to the same answer, so I'm confident that 196 has 9 divisors.

Final Answer The number of positive whole-number divisors of 196 is 9.
`</think>`

---

### ConPress-R1-7B

**INPUT:** How many positive whole-number divisors does 196 have?

**OUTPUT:** `<think>`

Alright, to find the number of positive divisors of 196, I should start by factoring it into its prime factors. I remember that 196 is a perfect square because $14 \times 14 = 196$. So, $196 = 14^2$. But 14 isn't prime; it's $2 \times 7$. So, substituting that back in, $196 = (2 \times 7)^2$. Using exponent rules, that's $2^2 \times 7^2$. Now, to find the number of divisors, I recall that if a number is expressed in its prime factorization form as

$$n = p_1^{a_1} \times p_2^{a_2} \times \cdots \times p_k^{a_k},$$

then the number of positive divisors is

$$(a_1 + 1)(a_2 + 1) \ldots (a_k + 1).$$

Applying that here, the exponents are 2 and 2 for primes 2 and 7, respectively. So, the number of divisors is $(2+1)(2+1) = 3 \times 3 = 9$.

Let me double-check that. The divisors of 196 should be: $1, 2, 4, 7, 14, 28, 49, 98, 196$. That's nine divisors. Yep, that seems right. `</think>`

