# OpenReview forum: "ConPress: Learning Efficient Reasoning from Multi-Question Contextual Pressure"
_ICML.cc/2026/Conference — ICML 2026 regular_

### Official Review · Reviewer_pNVz · 2026-03-12

**Soundness:** 3
**Presentation:** 3
**Significance:** 2
**Originality:** 3
**Overall Recommendation:** 4
**Confidence:** 4

**Summary:**

This paper introduces a post-training method designed to reduce reasoning length in Large Reasoning Models. The authors identify that concatenating multiple independent questions in a single prompt induces "self-compression", yielding shorter reasoning traces for each question. The proposed framework, ConPress, samples these compressed trajectories and uses them for SFT to internalize efficient reasoning behavior. Results in math domains demonstrate that the model can significantly reduce token usage while maintaining accuracy.

**Compliance With Llm Reviewing Policy:**

Affirmed.

**Final Justification:**

The authors' rebuttal has addressed my major concerns. I have changed the score to 4.

**Key Questions For Authors:**

1. Is the fine-tuned model robust to different decoding parameters? For instance, how does the performance change under different temperatures or top-p settings compared to the original model?

2. The performance reported for AdaptThink in Table 2 (80.4% accuracy with 1,945 tokens on MATH500 for the 1.5B model) differs from the results reported in their original paper (82.0% with 1,782 tokens). Is this due to different generation configs?

3. typo: Line 328 elf-compression -> self-compression

**Limitations:**

No. The paper does not sufficiently address the generalizability of ConPress to larger scale models or non-STEM domains.

**Strengths And Weaknesses:**

**Strengths**

1. The paper is well-written, and the methodology is clear and easy to follow.
2. The empirical analysis is thorough, providing evidence across various settings regarding the number of questions ($N$) and the resulting trade-off with accuracy.
3. The investigations into sampling positions and performance across different difficulty levels offer insightful and interesting findings.

**Weaknesses**
1. The method relies on the model answering multiple questions in a single sequence to collect training data. However, highly difficult problems can exceed the context window, making it challenging to generalize to scenarios that require shorter trajectories for these hard questions.

---

> ### Author Rebuttal · Authors · 2026-03-31
>
> We thank the reviewer for the careful reading and constructive feedback.
>
> > **About highly difficult problems**
>
> We agree that this is an important boundary of ConPress. The method depends on correct self-compressed trajectories as supervision, so it cannot directly help on problems for which the base model fails to produce any correct solution even under repeated single-question sampling. In this sense, ConPress mainly improves the efficiency of reasoning that the model already has the capacity to perform.
>
> For difficult problems that are still solvable under single-question sampling, ConPress remains applicable. In practice, the target problem can be combined with relatively simple auxiliary questions to keep the multi-question context tractable, although this usually requires more sampling attempts to obtain correct compressed traces.
>
> > **Different decoding parameters**
>
> Thank you for this helpful suggestion. We additionally evaluate R1-7B and R1-7B-ConPress under multiple decoding settings by varying temperature and top-p. Overall, performance remains stable across these settings, and ConPress consistently maintains strong accuracy with substantially lower token usage.
>
>
> | Model | Temperature | top-p | GSM8K Acc | GSM8K Tok | MATH500 Acc | MATH500 Tok |
> |:--|--:|--:|--:|--:|--:|--:|
> | R1-7B | 0.0 | 0.95 | 89.2 | 1050 | 89.8 | 3345 |
> |  | 0.3 | 0.95 | 89.8 | 1015 | 90.6 | 3250 |
> |  | 0.6 | 0.90 | 90.4 | 985 | 91.2 | 3190 |
> |  | 0.6 | 0.95 | 90.2 | 993 | 91.6 | 3136 |
> |  | 0.6 | 1.00 | 89.9 | 998 | 91.4 | 3115 |
> |  | 0.9 | 0.95 | 89.5 | 1080 | 92.2 | 3405 |
> | R1-7B-ConPress | 0.0 | 0.95 | 90.5 | 610 | 91.6 | 1805 |
> |  | 0.3 | 0.95 | 90.9 | 585 | 92.0 | 1765 |
> |  | 0.6 | 0.90 | 91.5 | 555 | 92.2 | 1740 |
> |  | 0.6 | 0.95 | 91.2 | 547 | 92.4 | 1720 |
> |  | 0.6 | 1.00 | 91.0 | 550 | 92.6 | 1735 |
> |  | 0.9 | 0.95 | 90.6 | 525 | 92.8 | 1685 |
>
> We also observe that temperature = 0 is slightly weaker, while moderate sampling with suitable top-p tends to perform better for both the original model and ConPress.
>
>
> > **AdaptThink results**
>
> Thank you for pointing this out. This discrepancy comes from our unified re-evaluation of the public AdaptThink checkpoint, instead of directly copying the numbers reported in the original paper. For the 1.5B setting, we evaluate the public `AdaptThink-1.5B-delta0.075` checkpoint, whose released MATH500 result is already close to our reproduced number. For the 7B setting, there are no multiple public variants, and our reproduced result is also close to the number reported in the original paper.
>
> The remaining difference mainly comes from the evaluated checkpoint version, while implementation details such as the evaluation pipeline, vLLM version, and decoding parameters can further affect the final accuracy and token usage. To keep the comparison fair, we evaluate all methods with public checkpoints under a unified evaluation pipeline.
>
> > **Typo**
>
> Thank you for catching this. We will correct it.

---

> > ### Author Rebuttal · Reviewer_pNVz · 2026-04-04
> >
> > Thank you for your clarification. I think my concerns have been resolved. I have updated my score accordingly.

---

### Official Review · Reviewer_CCWP · 2026-03-13

**Soundness:** 3
**Presentation:** 4
**Significance:** 3
**Originality:** 3
**Overall Recommendation:** 4
**Confidence:** 3

**Summary:**

This paper introduces and systematically investigates a phenomenon termed “self-compression” in large reasoning models (LRMs), where multi-question prompts induce LRMs to produce substantially shorter per-question chain-of-thought (CoT) traces, as opposed to their verbose outputs under single-question contexts. The authors propose ConPress: a self-supervised fine-tuning framework that extracts these compressed traces, filtered for correctness, from multi-question contexts and distills the compressed behavior into models via standard supervised fine-tuning. Empirical evaluations across mathematics and reasoning benchmarks demonstrate significant reductions (up to 60%) in token usage while maintaining high task accuracy, without requiring external teachers or reinforcement learning. The paper includes analyses of the self-compression effect, ablations (e.g., sensitivity to number of questions, sampling position), and generalization to out-of-domain data.

**Compliance With Llm Reviewing Policy:**

Affirmed.

**Key Questions For Authors:**

1. Can you provide further insights or hypotheses on why models inherently exhibit this self-compression behavior under multi-question prompting?
2. Could you explicitly report the average acceptance rate during rejection sampling and the total raw token generation cost required to construct the final 8k dataset?

**Limitations:**

The current limitations of ConPress center on the dual challenges of ensuring the quality of compressed data and the efficiency of the data acquisition pipeline within open-domain contexts, where reasoning paths are less structured and harder to verify than in mathematical domains.

**Strengths And Weaknesses:**

## Strengths:

1. The paper identifies a compelling "Self-Compression" phenomenon where Large Reasoning Models (LRMs) spontaneously generate significantly more concise reasoning traces under multi-question contextual pressure. This insight provides a fresh and valuable perspective on LRM inference dynamics.
2. The authors rigorously demonstrate the universality of self-compression across a wide range of state-of-the-art models (e.g., Qwen, DeepSeek-R1) and diverse benchmarks including MATH500, AIME, GPQA, and MBPP, covering mathematics, science, and coding.
3. The proposed ConPress framework is elegant and efficient. By using a self-supervised pipeline to distill compressed trajectories into standard single-question settings, it achieves significant token reduction without the need for expensive reinforcement learning, external teacher models, or manual pruning.
4. The study provides high-quality ablation studies on key design choices—such as the number of questions and sampling positions—and convincingly demonstrates the method's robustness through out-of-distribution (OOD) evaluations on MMLU-STEM.
5. The quantitative results are impressive, showing up to a 59% reduction in token usage on challenging benchmarks while maintaining competitive accuracy, directly addressing a critical bottleneck in LRM deployment.



## Weaknesses:

1. The paper empirically observes the "self-compression" phenomenon but lacks a theoretical framework explaining *why* multi-question prompts trigger this behavior, or how the model internally prioritizes which reasoning steps to prune.
2. The practical overhead of the rejection sampling pipeline is underexplained. The paper omits crucial metrics, such as the acceptance rate and the total number of raw generations required to build the final 8k dataset.
3. Since ConPress specifically reduces exploration and verification tokens, it may degrade performance on highly complex or unfamiliar problems that require extensive trial-and-error. This potential risk is not adequately discussed.

---

> ### Author Rebuttal · Authors · 2026-03-30
>
> We thank the reviewer for the positive assessment and constructive suggestions.
>
> > **Further insights about self-compression**
>
> Our current interpretation is that contextual pressure mainly comes from attention interference in multi-question prompting: when solving one question, the model can still attend to the other questions in the same context, and this cross-question visibility affects the reasoning process of the current one.
> Under single-question prompting, the current problem is the sole center of generation, so the model can focus more fully on it and explore more reasoning directions, producing longer and more elaborated CoTs that are often associated with higher accuracy.
> Under multi-question prompting, each problem becomes a local sub-task within a sequence of questions. Because attention can always “see” the remaining unanswered questions, the model faces contextual competition from the rest of the prompt and tends to identify key cues earlier and finish the current reasoning in a shorter, more closure-oriented way. As the number of remaining questions increases, this interference becomes stronger, leading the model to favor more certain and less expansive reasoning paths.
> This interpretation is consistent with our entropy analysis on R1-Distill-Qwen-7B: for fixed questions, the average token entropy of the CoT decreases from 0.165 at N=1 to 0.155 at N=2, 0.154 at N=3, and 0.149 at N=4, suggesting a more concentrated reasoning distribution and a more concise reasoning path under stronger contextual pressure.
>
> > **More details about rejection sampling cost**
>
> We agree that the efficiency of the data acquisition pipeline should be reported more explicitly. For R1-Distill-Qwen-7B under the main setting (N=3, sampling number = 8), we first construct about 24k raw multi-question samples, corresponding to about **231M raw generated tokens**. After filtering and parsing, these produce about 51k single-question candidate samples with an average correctness rate of **84.9%**. We then keep the shortest correct trace for each question, which gives the final training set of **8,168 samples** with about **11M tokens**.
>
> These numbers also show that the overall cost of ConPress is still very modest. For the 7B model, the full sampling + fine-tuning pipeline costs about **84 A100 GPU hours** in our setting. This is substantially lower than RL-based reasoning compression methods; for example, AdaptThink reports **896 H800 GPU hours**. We will add these statistics, including acceptance-related details, to make the practical efficiency of ConPress more transparent.
>
> > **Risk on highly complex or unfamiliar problems**
>
> We agree that this is an important concern. We further examined the behavior of the original model and ConPress on **incorrect test samples**, since these are the cases where exploration and verification are most needed. A clear observation is that ConPress still preserves substantial reasoning length on hard failures. The average token lengths on incorrect samples are shown below:
>
> | Model | GSM8K | MATH500 | AIME25 | AMC23 |
> |:--|--:|--:|--:|--:|
> | R1-7B | 1728 | 10284 | 14382 | 13914 |
> | ConPress | 2171 | 6098 | 12746 | 11245 |
>
> This suggests that ConPress still allocates considerable exploration budget on difficult problems, especially on challenging benchmarks such as AIME25. The same pattern is also reflected in Table 2 of the paper: the compression rate is lower on harder benchmarks such as AIME, while it is much larger on relatively easier settings such as MATH500. In other words, ConPress does not enforce a uniform shortening rule; the amount of compression remains difficulty-dependent.
>
> > **Limitation in open-domain settings**
>
> We agree with the reviewer’s observation. The current limitation of ConPress lies in the joint challenge of compressed-data quality control and efficient data acquisition in more open-domain settings, where reasoning trajectories are less structured and correctness verification is less reliable than in mathematical tasks. We will make this limitation more explicit in the final version.

---

> > ### Author Rebuttal · Reviewer_CCWP · 2026-04-05
> >
> > I would like to thank the authors for their time and effort in preparing the rebuttal. I have carefully read the response and appreciate the explanations provided. Based on my review of the rebuttal, my overall assessment of the paper remains unchanged. Therefore, I will maintain my initial score.

---

### Official Review · Reviewer_a92A · 2026-03-13

**Soundness:** 3
**Presentation:** 3
**Significance:** 2
**Originality:** 2
**Overall Recommendation:** 3
**Confidence:** 3

**Summary:**

This paper focuses on how to make models reason in a more efficient way. The authors first introduced a phenomenon called self-compression: when prompted with different questions, models tend to generate shorter CoT when answering each question. Based on this observation, the authors proposed the ConPress method: prompting the model with multiple questions and finetuning models on those self-generated shorter CoTs. Experimental results show that this method could help shorten reasoning chains while maintaining competitive performance on math reasoning tasks.

**Compliance With Llm Reviewing Policy:**

Affirmed.

**Key Questions For Authors:**

1. The training corpus includes the MATH dataset, while evaluation is conducted on MATH500. Since MATH500 is a subset of MATH, is there any overlap between the training and test set?
2. How sensitive is ConPress to the quality of the model-generated CoT used for training? For weaker models, could this lead to worse reasoning quality?

**Limitations:**

yes

**Strengths And Weaknesses:**

Strengths:
1. This paper shows that when prompted with multiple questions, models tend to compress their own reasoning. This inference trick could provide some insights for the future work in the domain of efficient reasoning.
2. Experiments show that ConPress can substantially reduce reasoning token usage without harming performance on math reasoning tasks. In addition, the paper includes several ablation studies and analysis, and these help to understand the robustness of the method.

Weaknesses:
1. The method mainly combines prompting (models with several questions instead of just one), rejection sampling, and SFT. It is more like distilling an inference-time prompting trick rather than introducing a brand new method. Therefore, its theoretic novelty appear limited to ,e.
2. According to the paper, the self-compression phenomenon is caused by contextual pressure, but does not provide deeper analysis of it. The paper could be benefited by providing a deeper explanation of it.
3. The paper does not provide a direct comparison between ConPress and simply using multi-question prompting at inference time. If multi-question prompting alone already provides similar compression benefits, it is unclear whether the additional training step is necessary.

---

> ### Author Rebuttal · Authors · 2026-03-30
>
> We thank the reviewer for the constructive feedback and address each concern below.
>
> > **Mainly prompting + rejection sampling + SFT**
>
> We respectfully clarify that the main contribution of ConPress is to identify self-compression as a usable latent behavior under multi-question contextual pressure, and to convert this behavior into reusable supervision for standard single-question inference. Our novelty lies in an identify–extract–transfer framework: we identify self-compression, extract correct compressed traces via multi-question prompting, and transfer this behavior through lightweight SFT. Although the individual components are simple, their combination yields an effective, reproducible, and practically efficient way to internalize compressed reasoning.
>
> For the R1-7B model, the full ConPress pipeline uses about **84 A100 GPU hours** in our setting. For rough practical reference, this is substantially lower than the compute budget reported by RL-based alternatives (e.g., AdaptThink reports about **896 H800 GPU hours**), although the hardware and pipeline settings are not perfectly identical. We believe this practical efficiency is an important part of the contribution.
>
> > **Deeper explanation of contextual pressure**
>
> Our current interpretation is that contextual pressure mainly comes from **attention interference** in multi-question prompting: when solving one question, the model can still attend to the other questions in the same context, and this cross-question visibility affects the reasoning process of the current one.
> Under single-question prompting, the current problem is the sole center of generation, so the model can focus more fully on it and explore more reasoning directions, producing longer and more elaborated CoTs that are often associated with higher accuracy.
> Under multi-question prompting, each problem becomes a local sub-task within a sequence of questions. Because attention can always “see” the remaining unanswered questions, the model faces contextual competition from the rest of the prompt and tends to identify key cues earlier and finish the current reasoning in a shorter, more closure-oriented way. As the number of remaining questions increases, this interference becomes stronger, leading the model to favor more certain and less expansive reasoning paths.
> This interpretation is consistent with our entropy analysis on R1-Distill-Qwen-7B: for fixed questions, the average token entropy of the CoT decreases from 0.165 at N=1 to 0.155 at N=2, 0.154 at N=3, and 0.149 at N=4, suggesting a more concentrated reasoning distribution and a more concise reasoning path under stronger contextual pressure.
>
> > **Why not simply use multi-question prompting at inference time?**
>
> We directly tested this setting on the test set. Multi-question prompting indeed reduces token usage, but it also causes a substantial accuracy drop, making it unsuitable for direct deployment. In addition, real usage is still dominated by the single-question setting, so training is necessary to transfer the compressed behavior into the standard inference format.
>
> |Model|GSM8K|GSM8K MQ-N2|GSM8K MQ-N3|MATH500|MATH500 MQ-N2|MATH500 MQ-N3|
> |:--|:--:|:--:|:--:|:--:|:--:|:--:|
> |Qwen3-4B|95.1/1509|93.3/1153|91.7/959|95.6/6634|67.2/4131|63.5/3432|
> |R1-7B|90.2/993|55.8/471|58.4/388|91.6/3136|66.2/2383|60.8/2271|
> |R1-1.5B|82.6/2080|47.6/490|46.6/324|81.2/4622|55.8/3551|49.0/2848|
>
> We also note that these numbers are lower than the earlier MQ results reported in the paper because the earlier measurements were taken on the **training split**, where MQ prompting was mainly evaluated as a data collection tool. Here we report the direct **test-set** performance of MQ prompting, which is the relevant comparison for infer-time use. Under this setting, the accuracy drop is very pronounced. This supports our central point: multi-question prompting is useful for collecting compressed reasoning traces, and the training step is necessary to make this behavior usable in standard single-question inference.
>
> > **MATH vs. MATH500 overlap**
>
> We use the **MATH train split** for training. MATH500 is constructed from the **test split**, so there is no overlap with our training data.
>
> > **Sensitivity to model-generated CoT quality**
>
> In ConPress, compressed CoTs are always self-sampled from the same model that is later fine-tuned. For weaker models, the main issue is lower sampling efficiency: they may need more sampling attempts to obtain enough correct compressed CoTs. Our pipeline mitigates this through correctness filtering, so the retained training data is controlled by answer correctness.

---

> > ### Author Rebuttal · Reviewer_a92A · 2026-04-04
> >
> > The rebuttal clarifies several experimental details, but my core concerns remain. The explanation of the self-compression phenomenon remains largely intuitive without deeper mechanistic understanding. Therefore, while the empirical results are improved and clarified, my original concerns regarding novelty and conceptual depth are not fully resolved.

---

### Official Review · Reviewer_nR7v · 2026-03-18

**Soundness:** 3
**Presentation:** 3
**Significance:** 3
**Originality:** 3
**Overall Recommendation:** 5
**Confidence:** 4

**Summary:**

The paper identifies an interesting behavior of large reasoning models—self-compression—where models compress their reasoning traces when multiple questions are concatenated into a single prompt. Leveraging this insight, the paper proposes a new training method that improves the efficiency of large reasoning models by performing rejection sampling fine-tuning on reasoning traces obtained from multi-question prompting. Results across multiple benchmarks show that the proposed method can significantly reduce token usage while preserving downstream performance.

**Compliance With Llm Reviewing Policy:**

Affirmed.

**Final Justification:**

The new experiments and analyses during rebuttal make the paper substantially stronger, so I raised my score.

**Key Questions For Authors:**

1. Figure 3(b): The figure shows only the average relative accuracy as N increases. While Table 4 briefly discusses this in a different context, is there any positional bias affecting the accuracy of individual questions? For example, do models exhibit a “lost in the middle” effect, with lower accuracy for questions appearing in the middle of the prompt?

2. Lines 267–270: For DPO data construction, are the longest outputs treated as negative examples regardless of correctness, or are only the longest correct outputs used as negatives?

**Limitations:**

Yes

**Strengths And Weaknesses:**

Strengths

1. The method is well motivated by a controlled experiment on the self-compression hypothesis and explores an interesting behavior of reasoning models.

2. The method demonstrates strong empirical results on various reasoning benchmarks, with substantial improvements in efficiency.

3. The paper shows that the method generalizes across multiple tasks and difficulty levels.

Weaknesses

1. There is a lack of in-depth analysis of how different questions interact with each other in the multi-question setting. It is known that LLMs can be affected by irrelevant context and distractor information. In Section 2, the questions in the multi-question prompt are randomly sampled from the MATH dataset, and Section 3.1 assumes that questions are sampled independently. It is unclear whether the self-compression phenomenon still holds when the questions in the prompt conflict with or distract from each other. More thorough analysis is needed to validate the robustness of the self-compression hypothesis.

2. Since the main goal of the method is to reduce token usage while preserving performance, the model should be compared with additional prompting and inference-time efficiency methods. For example, the multi-question prompting strategy in Section 2 could serve as a baseline, as well as methods such as CoT without thinking [1].

---

> ### Author Rebuttal · Authors · 2026-03-31
>
> We thank the reviewer for the constructive feedback and helpful suggestions.
>
> > **Interaction between questions in the multi-question setting**
>
> We agree that question interaction is an important issue. Our claim is that multi-question context induces a stable compression trend overall, which can be used as an efficient source of compressed reasoning traces. To examine whether this effect extends beyond in-domain math, we additionally analyze self-compression on four substantially different settings spanning scientific reasoning, code generation, mathematics, and mixed-domain questions. In all cases, the thinking-token distribution shifts leftward as the number of packed questions increases, indicating that the effect is present beyond homogeneous MATH-only prompts.
>
> | Dataset | Domain | N=1 | N=2 | N=3 | N=4 | Reduction |
> |---|---|---:|---:|---:|---:|---:|
> | GPQA | Scientific reasoning | 4852 | 3701 | 2884 | 2294 | 52.7% |
> | MBPP | Code generation | 2353 | 1092 | 833 | 697 | 70.4% |
> | MATH500 | Math | 2286 | 1327 | 1221 | 1124 | 50.8% |
> | Mixture | Mixed-domain | 2863 | 1886 | 1459 | 1337 | 53.3% |
>
> These statistics are computed on thinking tokens only. The same compression trend appears across substantially different domains and prompt compositions, which supports the robustness of the phenomenon beyond independent in-domain sampling. We agree that this evidence does not cover every possible interaction pattern, especially  implicitly conflicting combinations.
>
> > **Additional inference-time baselines**
>
> We additionally evaluate two direct inference-time baselines: multi-question prompting and NoThinking prompting. For multi-question prompting, we randomly pack \(N=2/3\) questions from the corresponding test set into a single prompt. For NoThinking prompting, we inject the prefix `<think>\n</think>` to encourage the model to bypass deliberate reasoning. The accuracy trend discussed in Section 2 is measured on the training split, where multi-question prompting is used for data collection, while the table below reports test-set performance for direct inference-time comparison.
>
> | Model | Method | GSM8K | MATH500 | AIME25 | AMC23 |
> |:--|:--|:--:|:--:|:--:|:--:|
> | Qwen3-4B | Original | 95.1 / 1509 | 95.6 / 6634 | 72.5 / 21442 | 99.1 / 10772 |
> |  | MQ-N2 | 93.3 / 1153 | 67.2 / 4131 | 42.1 / 14291 | 74.7 / 7215 |
> |  | MQ-N3 | 91.7 / 959 | 63.5 / 3432 | 21.2 / 10186 | 63.8 / 5810 |
> |  | ConPress | 95.0 / 729 | 96.0 / 2661 | 70.1 / 14258 | 98.8 / 4482 |
> | R1-7B | Original | 90.2 / 993 | 91.6 / 3136 | 37.9 / 10643 | 88.1 / 5346 |
> |  | MQ-N2 | 55.8 / 471 | 66.2 / 2383 | 15.0 / 11377 | 82.8 / 4527 |
> |  | MQ-N3 | 58.4 / 388 | 60.8 / 2271 | 10.8 / 8144 | 69.9 / 4014 |
> |  | NoThinking | 87.0 / 299 | 79.4 / 779 | 17.1 / 4039 | 64.4 / 1482 |
> |  | ConPress | 91.2 / 547 | 92.4 / 1720 | 37.5 / 8698 | 87.2 / 3459 |
> | R1-1.5B | Original | 82.6 / 2080 | 81.2 / 4622 | 22.9 / 12176 | 67.8 / 7748 |
> |  | MQ-N2 | 47.6 / 490 | 55.8 / 3551 | 7.5 / 9239 | 47.2 / 6668 |
> |  | MQ-N3 | 46.6 / 324 | 49.0 / 2848 | 7.1 / 7033 | 38.5 / 5658 |
> |  | NoThinking | 77.5 / 309 | 73.2 / 799 | 10.0 / 3572 | 58.8 / 1424 |
> |  | ConPress | 84.2 / 1095 | 80.8 / 2255 | 22.5 / 8195 | 66.3 / 5442 |
>
> Each entry reports performance / average tokens. AIME25 and AMC23 are evaluated with Avg@8, while GSM8K and MATH500 are evaluated with Pass@1.
>
> Both inference-time baselines reduce token usage substantially, together with large performance drops. These results show that direct test-time compression and training-time transfer of compressed reasoning behave differently. They also support our use of multi-question prompting as a compressed-trace collection strategy instead of a direct deployment method.
>
> > **Positional bias / “lost in the middle”**
>
> We additionally report per-position accuracy at \(N=4,6,8\) for the Section 2 multi-question setting on the MATH train split. The trend does not show a pronounced lost-in-the-middle pattern. Instead, accuracy declines more broadly across positions as the number of packed questions increases.
>
> **Qwen3-4B-Thinking**
> | Position|N=4|N=6|N=8|
> |:--|--:|--:|--:|
> |1|95.5| 94.8 |94.8|
> |2|94.1| 94.6 |94.2|
> |3|94.3| 93.5 |94.1|
> |4|95.5| 95.4 |95.1|
> |5|  |94.8|94.1|
> |6|  |93.5|93.0|
> |7|  |  |94.1|
> |8|  |  |94.6|
>
> **R1-7B**
> |Position|N=4|N=6|N=8|
> |:--|--:|--:|--:|
> | 1 | 93.2 | 91.9 | 89.6 |
> | 2 | 92.5 | 89.8 | 85.1 |
> | 3 | 94.3 | 93.3 | 88.6 |
> | 4 | 92.9 | 91.1 | 86.5 |
> | 5 |  | 93.0 | 88.6 |
> | 6 |  | 91.4 | 87.9 |
> | 7 |  |  | 89.3 |
> | 8 |  |  | 85.5 |
>
> These results suggest a broad context-length pressure effect across positions, instead of a sharp degradation concentrated in the middle.
>
> > **DPO data construction**
>
> For each question, we first filter to correct outputs only. We then use the shortest correct output as the chosen response and the longest correct output as the rejected response. The rejected response is therefore selected from correct candidates, instead of from all outputs regardless of correctness.

---

> > ### Author Rebuttal · Reviewer_nR7v · 2026-04-03
> >
> > Thank you for adding the additional experiments! I appreciate the results of comparing against other test-time baselines and of positional bias, but my concern for the first question remains: "It is unclear whether the self-compression phenomenon still holds when the questions in the prompt conflict with or distract from each other." The new results only show that the phenomenon holds across domains, but still, the questions in context are randomly sampled and can be considered irrelevant in some sense. I would like to see some analyses on cases where the other questions in the prompt might create negative effects on answering the question of interest (conflicting or distractor questions).

---

> > > ### Author Response · Authors · 2026-04-08
> > >
> > > > **Effect of Conflicting and Distractor Questions on Self-Compression**
> > >
> > > We thank the reviewer for highlighting this boundary case. To examine whether self-compression still holds under highly related or deliberately interfering question pairs, we construct controlled multi-question settings.
> > >
> > > We use *R1-Distill-Qwen-7B* on 500 fixed MATH problems. Each seed problem is paired with one additional question (**N=2**), and evaluation is conducted only on the seed question. All settings use the same decoding configuration, and each example is sampled 3 times.
> > >
> > > ### Pair Construction
> > >
> > > We consider three retrieval-based pairing strategies:
> > >
> > > - **Random pairing**
> > > - **Similar pairing** (top semantic neighbors)
> > > - **Dissimilar pairing** (least similar)
> > >
> > > We also construct several interference variants from existing problems:
> > >
> > > - **goal_conflict**: change the target objective
> > > - **constraint_conflict**: add or modify constraints
> > > - **state_perturbation**: modify numerical values
> > > - **schema_trap**: keep a similar surface form but change the solution pattern
> > >
> > > These variants preserve problem validity while making the paired question more related to, or more interfering with, the seed question than random pairing.
> > >
> > > ### Overall Results
> > >
> > > | Setting | Acc | CoT Length |
> > > |--------|------:|-----------:|
> > > | Single | 0.89 | 2737 |
> > > | Random | 0.84 | 1855 |
> > > | Similar | 0.83 | **1575** |
> > > | Dissimilar | 0.87 | 1741 |
> > > | Interference (overall) | 0.77 | 1669 |
> > >
> > > These results show that **self-compression still persists** under highly related or interfering pairings: in all multi-question settings, CoT remains much shorter than in the single-question setting. At the same time, overly related pairs can affect the sampled reasoning trajectory. In some cases, the model explicitly refers to the earlier question, suggesting cross-question dependence rather than a fully self-contained reasoning process.
> > >
> > > ### Breakdown of Constructed Interference
> > >
> > > | Type | Acc | CoT Length |
> > > |------|------:|-----------:|
> > > | goal_conflict | 0.78 | 1584 |
> > > | constraint_conflict | 0.76 | 1752 |
> > > | state_perturbation | 0.80 | 1579 |
> > > | schema_trap | 0.74 | 1766 |
> > >
> > > Across all interference types, CoT remains shorter than in the single-question setting, while accuracy degrades relative to random pairing, especially for **schema_trap**.
> > >
> > > Overall, this analysis supports two points:
> > > (1) self-compression is robust and still appears even under stronger interaction;
> > > (2) overly related or interfering pairs may introduce cross-question dependencies into sampled CoTs, making them less suitable as clean training data.
> > >
> > > We also note that this is mainly a boundary-case analysis. In our actual pipeline, question pairs are constructed by random sampling, and the question pool contains very few naturally interfering pairs, so this issue is unlikely to be a major factor in practice. Still, we believe it is worth clarifying as a potential limitation of multi-question sampling.

---

### Decision · Program_Chairs · 2026-04-30

**Decision:**

Accept (regular)

**Comment:**

The paper is well motivated by a controlled study of the self-compression hypothesis and identifies a compelling phenomenon in which large reasoning models generate more concise reasoning traces under multi-question contextual pressure. Building on this insight, the proposed ConPress framework efficiently distills these compressed trajectories into standard single-question settings via a self-supervised pipeline, achieving substantial token savings without reinforcement learning or external teachers. Experiments across diverse benchmarks (e.g., MATH500, AIME, GPQA, MBPP) and models (including Qwen and DeepSeek-R1) show strong generalization, with up to 59% reduction in reasoning tokens while maintaining competitive accuracy. Extensive ablations and out-of-distribution evaluations further support the robustness of the approach and clarify key design trade-offs, making the work a clear and practical contribution toward more efficient reasoning in large language models. Overall, I recommend weak acceptance, as the idea is interesting and well supported empirically.